# Novel Perspectives in Chronic Kidney Disease-Specific Cardiovascular Disease

**DOI:** 10.3390/ijms25052658

**Published:** 2024-02-24

**Authors:** Cuicui Xu, George Tsihlis, Katrina Chau, Katie Trinh, Natasha M. Rogers, Sohel M. Julovi

**Affiliations:** 1Kidney Injury Group, Centre for Transplant and Renal Research, Westmead Institute for Medical Research, 176 Hawkesbury Road, Westmead, NSW 2145, Australia; cuicui.xu@wimr.org.au (C.X.); klam8836@uni.sydney.edu.au (K.T.); 2Renal and Transplantation Medicine, Westmead Hospital, Westmead, NSW 2145, Australia; george.tsihlis@health.nsw.gov.au; 3Department of Renal Services, Blacktown Hospital, Blacktown, NSW 2148, Australia; katrina.chau@health.nsw.gov.au; 4Blacktown Clinical School, School of Medicine, Western Sydney University, Sydney, NSW 2148, Australia; 5Faculty of Medicine and Health, The University of Sydney, Science Rd., Camperdown, NSW 2050, Australia

**Keywords:** Chronic kidney disease, cardiovascular disease, cardiorenal syndrome, inflammation, biomarkers

## Abstract

Chronic kidney disease (CKD) affects > 10% of the global adult population and significantly increases the risk of cardiovascular disease (CVD), which remains the leading cause of death in this population. The development and progression of CVD—compared to the general population—is premature and accelerated, manifesting as coronary artery disease, heart failure, arrhythmias, and sudden cardiac death. CKD and CV disease combine to cause multimorbid cardiorenal syndrome (CRS) due to contributions from shared risk factors, including systolic hypertension, diabetes mellitus, obesity, and dyslipidemia. Additional neurohormonal activation, innate immunity, and inflammation contribute to progressive cardiac and renal deterioration, reflecting the strong bidirectional interaction between these organ systems. A shared molecular pathophysiology—including inflammation, oxidative stress, senescence, and hemodynamic fluctuations characterise all types of CRS. This review highlights the evolving paradigm and recent advances in our understanding of the molecular biology of CRS, outlining the potential for disease-specific therapies and biomarker disease detection.

## 1. Introduction

Chronic kidney disease (CKD) is a major public health concern affecting at least 10% of the population worldwide (>800 million people), and this prevalence is expected to increase as a result of growing rates of diabetes mellitus and hypertension [1]. In 2017, CKD resulted in 1.2 million deaths and incurred a burden of 35 million disability-adjusted life years internationally [2]. Health resource allocation for CKD treatment has increased substantially in recent years in some countries; however, patients globally continue to experience dramatically reduced life expectancy. Morbidity and mortality among patients with CKD is largely attributable to cardiovascular disease (CVD) [3], with a markedly increased risk of cardiovascular events in adult and paediatric populations [4]. Cardiovascular mortality risk escalates with the severity of CKD, reaching 100× the rate in the general population [5]. Even mild-to-moderate loss of renal function is strongly associated with increased cardiovascular mortality [5,6]. Conversely, cardiovascular disease accounts for the highest annual healthcare expenditure (AUD 5.7 billion in Australia) [7]; mortality and medical costs increase with CKD severity.

Kidney and heart function are intrinsically linked, and this interdependent relationship—where the dysfunction of one organ induces pathological changes in the other—is known as cardiorenal syndrome (CRS). Despite a consensus definition published in 2010 [8], there has been minimal progress on our understanding of CRS pathophysiology [9], the exclusion of CRS patients from clinical trials [10], no biomarker development for clinical use [11], and limited therapeutic interventions that improve outcomes. CVD and CKD share modifiable co-morbidities (smoking, hypertension, and dyslipidaemia), non-modifiable comorbidities (increased age, genetic predisposition, and diabetes) and pathophysiology (systemic inflammation, oxidative stress, matrix deposition, and fibrosis) that contribute to the development and progression of disease. Understanding the pathogenesis of CRS requires focus on the molecular links that regulate cross-talk between the heart and kidney. Accumulated toxins are the cause of uraemia in CKD, and robust evidence establishes uraemic toxins as central to the deranged molecular pathways in both CVD and CKD [12]. However, protein-bound uraemic toxins are not removed by dialysis, and this poses management difficulties and a substantial therapeutic challenge.

## 2. Clinical Manifestations of CRS

The relationship between renal and cardiac function was first described by Robert Bright [13], but it took >150 years before a formal cardio-centric definition was formed [14], followed by bidirectional classifications [15] of clinical syndromes (namely CRS type 1–5) that recognized the primary contribution of both organs to acute or chronic pathophysiology. Although the Acute Dialysis Quality Initiative improved the definition of CRS phenotypes, the simplification is not necessarily utilitarian in clinical practice, where the initial insult and subsequent exacerbating events can be difficult to dissect and may not fundamentally change therapeutic decision making. A subsequent classification that is based on pathophysiological changes [16]—including hemodynamic, neurohumoral, inflammatory, and biochemical changes—broadly identifies causative factors that can direct the development of specific treatment options.

This review provides a predominantly “renocentric” view of CRS—there are many excellent reviews of primary cardiac dysfunction leading to renal impairment [11]. CVD manifestations require the dysregulation of different cell types within the heart (Figure 1). Ischemic heart disease due to coronary artery atherosclerosis is more prevalent in patients with CKD [17], demonstrating accelerated progression. Although there is no epidemiological relationship between the duration of renal replacement therapy, specifically haemodialysis, and the worsening of coronary artery disease [18], the incidence of myocardial infarction is high in the first week [19] and year [20] after commencing HD with uniformly poor outcomes. A recent meta-analysis demonstrated fewer cardiovascular events in patients undergoing peritoneal dialysis, but mortality was greater [21]. The causality underscoring these differences is not well understood, although haemodialysis-specific syndromes such as intradialytic hypotension and myocardial stunning are contributors [22,23]. More recent data suggest that the aggressive management of traditional CV risk factors has led to a decline in the incidence of significant coronary artery disease [24].

### 2.1. Coronary Artery Disease

While the total atheromatous burden is similar in CKD (compared to the general population), the nature of coronary artery plaque is altered. The most marked difference is hydroxy-apatite deposition leading to intimal and medial calcification and medial wall thickening [25,26]. The clinical presentation of CAD is also modified in CKD patients, often with a paucity of symptoms limited to dyspnoea and fatigue, or arrythmias, and lacking classical chest/arm pain. The risk of impaired coronary perfusion is exacerbated by concurrent left ventricular hypertrophy (LVH), which increases myocardial oxygen demand. Recurrent occult ischemia promotes cardiomyocyte apoptosis and extracellular matrix (ECM) deposition, leading to fibrosis (which, in turn, promotes capillary rarefaction) [27], LV stiffness, and diastolic dysfunction. The supply–demand mismatch predisposes patients to non-ST-segment elevation myocardial infarction, although presentation with acute ST-elevation infarction is more common [28]. Coronary artery calcification serves as a robust marker of CV risk in CKD patients [29], but evidence on modifiability remains limited [30].

### 2.2. Left Ventricular Hypertrophy (LVH)

The prevalence of LVH is estimated to be 30% in CKD patients, increasing to 80% with end-stage disease, and remains a strong independent predictor of survival. Increases in afterload (systolic hypertension and arterial remodelling that promotes stiffness), preload (intravascular volume expansion), and non-traditional risk factors (discussed below) contribute to initial concentric hypertrophy and heart failure with preserved ejection fraction (HFpEF). The disproportionate rise in LV diastolic pressure for any increment in volume leads to retrograde pulmonary venous congestion, eventually impacting right heart function. An elevated central venous pressure can further impair renal function by limiting the glomerular perfusion gradient. An ongoing LV overload promotes further maladaptive changes, including eccentric hypertrophy and dilatation, resulting in HF with reduced EF (HfrEF). The risk of hospitalization, CV death, and all-cause mortality in patients with congestive heart failure is increased in CKD [31]. More recent data suggest that CKD has a greater impact on mortality with an HfrEF phenotype [32], as well as in patients undergoing percutaneous coronary intervention [33]. The bidirectional impact of HF and CKD has been seen in the placebo groups of recent notable cardiovascular trials, including EMPEROR [34], DAPA-HF [35], DAPA-CKD [36,37], and CREDENCE [38].

### 2.3. Valvular Heart Disease

Valvular heart disease is prevalent in patients with CKD, with a predilection for aortic and mitral valves that contributes to stenosis and regurgitation phenotypes [39]. Aortic valvular disease is most common, and valve calcification is likely accelerated by deranged calcium–phosphate balance and metabolic bone disease driven by secondary hyperparathyroidism, shear stress endothelial damage, and the accumulation of uremic toxins. Both native and bioprosthetic valves are susceptible to accelerated structural deterioration in CKD compared to the general population [40].

### 2.4. Sudden Cardiac Death

Sudden cardiac death is a common cause of CV death in CKD patients undergoing dialysis-based renal replacement therapy [5], with an estimated event rate of 59 deaths per 1000 patient years (compared to 1 death per 1000 patient years in the general population). Data from the USRDS demonstrates that the rate of sudden cardiac death is 50% higher in HD compared to PD in the first 3 months after commencing RRT, although the difference does not persist beyond 2 years [41]. It is thought that the intermittent nature of haemodialysis—particularly the electrolyte derangement and volume shifts that are the most marked in the immediate post-dialysis and dialysis-free intervals—rather than coronary artery disease per se may be a significant contributor [42,43]. Few studies have documented causative arrhythmias at the time of sudden cardiac death, with the majority being ventricular fibrillation/tachycardia [44] in addition to asystole and bradyarrhythmia [45]. The use of implantable cardioverter defibrillator devices is effective in the general population, but using them in end-stage kidney disease did not reduce the rate of sudden cardiac death or all-cause mortality [46,47].

## 3. Molecular Mechanisms Underlying CRS

There are very few pre-clinical or human studies that focus specifically on cardiorenal syndrome (CRS); therefore, many of the molecular mechanisms identified are extrapolated from the existing cardiovascular literature in the context of normal renal function. Until recently, many large cardiovascular trials excluded patients with any degree of impaired renal function. Despite the robust association between CKD and cardiovascular morbidity and mortality, CRS patients are excluded from approximately 80% of clinical trials [48] and insufficiently considered in guidelines, resulting in a paucity of published data. In rodent studies—regardless of the mechanism of the primary renal insult—cardiovascular manifestations (hypertension, LVH, etc.) can vary by genetic background [49] as well as other factors including microbiome and age. In addition, the traditional 5/6 nephrectomy model, which is considered an excellent translational model to compare with human disease [50], is hampered by high mortality [51] and is less frequently used. This highlights the fact that molecular pathways driving CRS—and potential therapeutic opportunities—remain to be discovered. Here, we summarise recent advances in the field of cardiorenal syndrome, including putative mechanisms that worsen the disease, and potential clinical implications.

## 4. Inflammation in CRS (Figure 2)

The role of inflammation in the progression of kidney diseases has been firmly established over the past three decades [52], and in pre-clinical studies, this appears to be independent of the primary renal insult. Chronic inflammation in patients with CKD, particularly those undergoing dialysis, is associated with a heightened risk of all-cause and CV mortality [53]. Individuals with CKD exhibit elevated levels of inflammatory markers, including C-reactive protein (CRP) [54], interleukin (IL)-1β, IL-6, and tumour necrosis factor (TNF) [55]. The inflammatory response in CKD is multifactorial, stemming from uremic toxins that promote oxidative stress [56], a predisposition to infection [57], intestinal dysbiosis [58], metabolic acidosis [59], and diminished renal clearance of cytokines and toxins [60].

**Figure 2 ijms-25-02658-f002:**
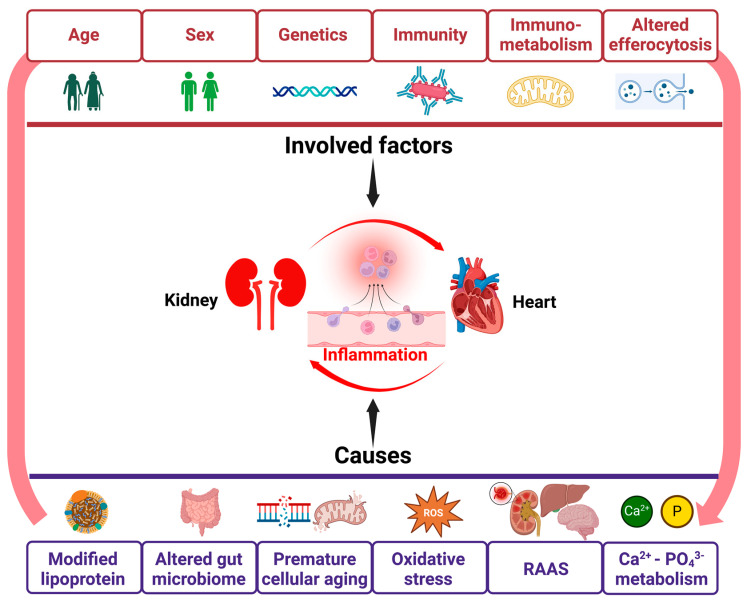
Factors involved in cardiorenal inflammation. The kidney and heart functions are interdependently linked, and several common factors have been involved in the chronic inflammation demonstrated in chronic kidney disease (CKD) and cardiovascular disease (CVD). Abbreviation: RAAS, renin angiotensin aldosterone systems.

A pro-inflammatory milieu attenuates the stimulatory effects of erythropoietin [61], elevates resting energy expenditure [62], and suppresses the production of growth hormone, insulin-like growth factor 1, and anabolic hormones (reviewed in [63]), leading to sarcopenia and malnutrition [64]. Systemic inflammation is associated with both HF [65] and acute CRS [66] in human studies, and the causal significance of inflammation in CV disease among patients with CKD is underscored by the Canakinumab Anti-Inflammatory Thrombosis Outcomes Study (CANTOS) trial [67], and a recent secondary analysis where only inflammatory biomarkers, and not LDL cholesterol, predicted CV events [68]. Similarly, a double-blind, randomized, placebo-controlled phase 2 trial of the IL-6 antibody ziltivekimab demonstrated reduced inflammatory and thrombotic markers in patients with stage 3/4 CKD at high risk of atherosclerotic events [69], reduced erythropoietic requirements in those with end-stage kidney disease (a recognised risk factor for CV disease), and promoted inflammatory hyporesponsiveness [70].

Biomarkers of inflammation, such as high-sensitivity CRP (hs-CRP), tumour necrosis factor-α receptor 2 (TNF-αR), white blood cell count, and IL-6, have been identified as predictors for the development of CKD [71]. In the Chronic Renal Insufficiency Cohort, inflammatory metrics, including hs-CRP, IL-1β, IL-1 receptor antagonist, IL-6, TNF-α, TNF-β, fibrinogen, and serum albumin, predicted faster progression to kidney failure [71] and atherosclerotic vascular disease and death, which has been confirmed by more recent studies [72,73,74]. However, a meta-analysis investigating the role of hs-CRP in coronary artery disease, stroke, and death in the general population failed to demonstrate any predictive relevance of this biomarker [75,76].

### 4.1. Genetics of Inflammation

In the largest genome-wide association study on CRP in UK Biobank participants (N = 427,367 people of European descent) and the Cohorts for Heart and Aging Research in Genomic Epidemiology (CHARGE) Consortium (total N = 575,531, also a predominantly European population), a weighted genetic risk score of CRP was associated with 27 clinical outcomes, including coronary heart disease [77]. However, the inverse variance weighting method failed to confirm the causal nature of this association.

In the general population, a missense variant in the IL-6 receptor gene locus (rs2228145) has been linked to lower levels of systemic inflammation and a reduced risk for myocardial infarction and stroke [78]. Similarly, a common intronic variant (rs10754555) of the NLRP3 gene with a 40% minor allele frequency has been associated with evidence of inflammasome activation, high hs-CRP, and an elevated risk of ischemic heart disease and cardiovascular mortality [79]. In European (Italian) patients with stage 2–5 CKD, high serum IL-6 was associated with a history of cardiovascular disease and predicted incident cardiovascular events, and this relationship was accompanied by the G174C polymorphism in the IL-6 gene [80]. This finding has been corroborated in other ethnicities [81]. Large-scale analyses from the UK Biobank have shown that genetic differences in IL-6 signalling preferentially modified the coronary artery disease risk to a greater extent among those with CKD than without [82].

Genetic studies in patients maintained on RRT also support a key role of inflammation—a recent study demonstrated that an IL-6 gain-of-function variant was significantly associated with the risk of all-cause mortality—although not specifically CV outcomes [83]. Genetic association studies of inflammation in patients with CKD are limited, although the application and integration of genomics to the diagnosis of CKD may change the current knowledge base. A small case–control study, including patients undergoing dialysis and healthy control subjects, found that single nucleotide polymorphisms within the Il1rn, Il1a, and Il1b genes were associated with a higher risk of end-stage kidney disease [84]. In a study involving >2000 Japanese individuals, the rs4845625 variant of the IL6 receptor was significantly associated with CKD [85].

### 4.2. The Impact of Innate Immune Signalling (Figure 3)

Inflammation is also key in the initiation, progression, and clinical manifestation of CVD. Recently published data from >30,000 patients undergoing statin therapy suggests that the residual inflammatory risk is more significantly linked to future CV events than LDL cholesterol [86]. The recognition of various molecular patterns with shared structural and physicochemical properties by pattern recognition receptors (PRRs) represents a fundamental principle of the innate immune system [87]. These patterns can originate from exogenous pathogens, such as bacteria, viruses, or fungi [known as pathogen-associated molecular patterns (PAMPs)], or from endogenous mediators released from damaged cells (known as damage-associated molecular patterns (DAMPs)) [88]. The activation of PRRs alters the activity of transcription factors, inducing the expression of pro-inflammatory cytokines (in the case of NF-κB and activator protein-1 (AP1)) or anti-inflammatory pathways (in the case of Nrf2) [89], reducing oxidative stress, inducing the expression of antioxidative enzymes such as catalase and superoxide dismutase, and directly regulating the expression of pro-inflammatory NF-κB target genes.

**Figure 3 ijms-25-02658-f003:**
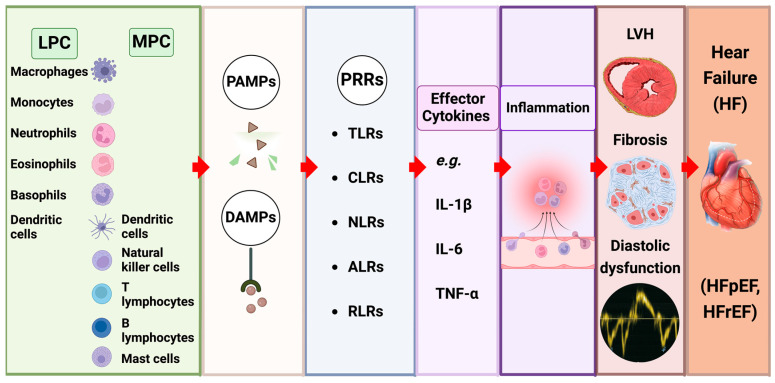
Inflammation underlies the molecular mechanisms in cardiorenal syndrome (CRS). The innate and adaptive immune system are composed of various immune cells that can respond to pathogen-associated molecular patterns (PAMPs) and damage-associated molecular patterns (DAMPs) through recognising cytosolic or surface pattern recognition receptors (PPRs). Effector responses include the release of cytokines, e.g., interleukin (IL)-1β, IL-6, and tumour necrosis factor (TNF)-α. The excess and/or prolonged existence of these cytokines can lead to the clinical manifestations of left ventricular hypertrophy (LVH), fibrosis, and diastolic dysfunction. With or without treatment, the disease can typically progress into heart failure (HF), which can be differentiated into HF with preserved ejection fraction (HFpEF) and/or HF with reduced EF (HFrEF).

Several classes of PRRs exist, including Toll-like receptors (TLRs) and C-type lectin receptors expressed on the cell surface, as well as NOD-like receptors (NLRs), absent in melanoma 2 (AIM2), and retinoic acid-inducible gene-I-like receptors (RIG I-like receptors) located in the cytoplasm. In response to distinct PAMPs and DAMPs, NLRs and AIM2 form multimeric cytosolic protein complexes, known as inflammasomes. Of these, the NLRP3 inflammasome has been most extensively studied in CRS. Two signals are activated by DAMPs during stress. The first is through the classical TLR2/4 pathway, releasing IL-1β. This cytokine is fundamental to maintaining cardiac alterations during CRS type 3 [90]. The second signal involves the NLRP3 pathway, initiated by the phagocytosis of DAMPs and activated through lysosomal damage. The activation of the NOD-type intracytoplasmic receptor (NLR) family leads to the formation of protein complexes called inflammasomes, including the NLRP3 inflammasome. NLRP3 is formed by three basic structures: NOD-type receptors, adapter protein (ASC), and pro-caspase 1. These active caspases induce the proteolytic maturation of IL-1β and IL-18, which are released into the extracellular environment to stimulate other cells, such as cardiomyocytes and podocytes [91]. Both signalling pathways have been studied in acute kidney injury [90] (renal ischemia reperfusion injury) and diabetic mice [92], and in both models, signal 2 demonstrated significant importance for maintaining cardiac alterations (via macrophages), suggesting it as a potential therapy tool for CRS.

### 4.3. Inflammatory Cytokines as Effector Molecules in CRS

The IL-1 family of cytokines consists of 11 cytokines (including IL-1α, IL-1β, and IL-18) that bind to 10 IL-1 receptors (IL-1Rs) [93] and are relevant as mediators and/or potential therapeutic targets in CVD [94]. While the maturation of IL-1β is mediated by the inflammasome, the processing of pro-IL-1α is inflammasome-independent [95]. In contrast to pro-IL-1β, pro-IL-1α is constitutively expressed by various cell types and is already biologically active in its proprotein form [96]. It can translocate into the nucleus, acting as a transcription factor, but can also be shuttled to the cell membrane, mediating the activation of other cells via the activation of IL-1R1 in a paracrine manner [97]. Intracellular pro-IL-1α is cleaved by the calcium-dependent protease calpain, and membrane-associated pro-IL-1α is cleaved by extracellular proteases, such as granzyme B [97,98]. Membrane-bound pro-IL-1α can also be processed by thrombin, establishing a link between innate immunity and coagulation, which is relevant in the context of wound healing, platelet loss, and human sepsis [99], as well as atherosclerotic plaque rupture and acute myocardial infarction.

Altered innate immunity has emerged as a factor in the pathogenesis of CVD from atherosclerosis to HF [100]. Every step of atherogenesis, from endothelial dysfunction, to transforming macrophages into foam cells, to plaque formation and rupture, depends on inflammatory cytokines and the cellular constituents of the inflammatory response. In myocardial infarction, hematopoietic stem and progenitor cells are activated with a shift towards myelopoiesis [101]. Psychological stress, a factor of major relevance for coronary heart disease [102], induces the remodelling of chromatin and the transcriptomic reprogramming of monocytes to a primed hyperinflammatory phenotype [103].

Interestingly, lipopolysaccharide injection in mice on a Western diet that are subsequently switched back to a normal diet incites a stronger pro-inflammatory response. This phenomenon—defined as “hematopoietic reprogramming”—is mediated by NLRP3 in the bone marrow [104]. Somatic mutations in the cells of the hematopoietic system accumulate during aging and are detected in about 10% of individuals over 65 years of age. These mutations arise in genes associated with leukemia or lymphoma and are defined as “clonal haematopoiesis of indeterminate potential” (CHIP) [105]. The risk of these mutations extends to CVD because carriers are at a higher risk of myocardial ischemia and ischemic stroke [106]. Further research is required to determine whether these pathways would be relevant in CKD and the development of CRS.

### 4.4. Innate Immune Cells in CRS

Cells of the monocyte/macrophage/dendritic cell lineage, as well as T cells, are crucial effectors of the innate immune response [107] that are pathologically involved in the development of CRS, although data are sparse. An angiotensin II (Ang II)-induced model of renal damage demonstrated that renal DC expressed major histocompatibility complex (MHC) class II and CD86 (markers of maturation), promoting the infiltration of CD4^+^ and CD8^+^ T cells into the kidney [108]. This effect was limited by the administration of the anti-lymphoproliferative agent mycophenolate mofetil, prednisolone, or the TNFa decoy receptor etanercept.

The coordination of the immune response following injury leads to sequential immune cell infiltration into the heart (as it does to the kidney), contributing to cardiac inflammation and fibrosis. Experimental models of HF suggest a pivotal role for pro-inflammatory mediators in development, progression, and mortality [109]. However, clinical trials directly targeting TNF-α in HF have shown no benefit [110,111], although more recent clinical trials targeting IL-6 [69] and IL-1b [67] have shown greater therapeutic promise. Thus, immune modulation in HF remains a subject of debate.

A recent study using a two-hit preclinical HF model demonstrates that impaired T cell IRE1α/XBP1 signalling directs inflammation in HfpEF [112]. Additionally, high sodium intake can impact systemic immune responses by altering the intestinal microbiome and immunomodulatory metabolites [113]. High sodium levels promote Th17 cell polarisation that display a pro-inflammatory phenotype [114] and provoke experimental autoimmune encephalitis. The potential link with CRS is unknown, but Th17 cells are linked with hypertension [115]. In mice, a high salt intake also depletes *Lactobacillus murinus*, which is known to limit gut dysbiosis and the production of uremic toxins, particularly indoxyl sulphate [116].

## 5. Contributors to Inflammation in CRS

A multitude of endogenous and exogenous mediators induce inflammation in both the general population and in patients with CKD. These include modified lipoproteins, alterations in the gut microbiome that facilitate the production of uremic toxins, factors associated with premature aging, and disturbances in calcium–phosphate metabolism.

### 5.1. Modified Lipoproteins

CKD amplifies the pro-inflammatory properties of various lipoproteins by inducing alterations in protein composition (proteome), lipid composition (lipidome), post-translational protein modifications, and the accumulation of small molecules within lipoprotein particles [117]. The carbamylation of LDL and HDL is highly prevalent in the lipoproteins of patients with CKD, enhancing pro-atherogenic properties by reducing the release of nitric oxide (NO) from endothelial cells and promoting endothelial dysfunction. The guanidinylation of triglyceride-rich lipoproteins, particularly apolipoprotein C3 (resulting in gApoC3), activates the NLRP3 inflammasome in human monocytes [118]. gApoC3 accumulates in the plasma of patients with CKD, demonstrating pro-inflammatory properties and associations with both kidney parenchymal fibrosis (pre-clinical model) and CV events (in humans) [119]. CKD also impairs the vasoprotective properties of HDL [120,121].

### 5.2. Altered Gut Microbiome and Disturbed Intestinal Barrier Function (Figure 4)

CKD is associated with the accumulation of toxins due to increased production, impaired intestinal barrier function, and reduced clearance [122]. These substances include a range of endotoxins and bacterial metabolic products [e.g., indoxyl sulphate, p-cresyl sulphate, trimethylamine N-oxide (TMAO), and phenylacetylglutamine] that can activate the innate immune system. For example, endotoxin interacts directly with TLRs on immune effector cells, leading to the NF-κB- and MAPK-dependent production of pro-inflammatory cytokines. Indoxyl sulphate and TMAO interfere with reverse cholesterol transport, increasing the accumulation of pro-atherogenic lipoproteins in macrophages [123,124], although these studies were not performed in a CKD population. Elevated plasma concentrations of these mediators are associated with endothelial dysfunction, CVD, CV events, and mortality in patients with [125] or without CKD [126] (and comprehensively reviewed in [12]). The aryl hydrocarbon receptor is another important intracellular target of uremic toxins (particularly indoxyl sulphate), as demonstrated in a recent study [127]. This study revealed that in the context of CKD, both indoxyl sulphate and the extracellular matrix protein thrombospondin-1 (TSP1) promote features of left ventricular hypertrophy (LVH) through the activation of the aryl hydrocarbon receptor (AhR). The blockade of TSP1 signalling by genetic or pharmacological disruption reduced the cardiomyocyte size, senescence-associated secretory phenotype (SASP—see below), and MAPK signalling in vitro.

**Figure 4 ijms-25-02658-f004:**
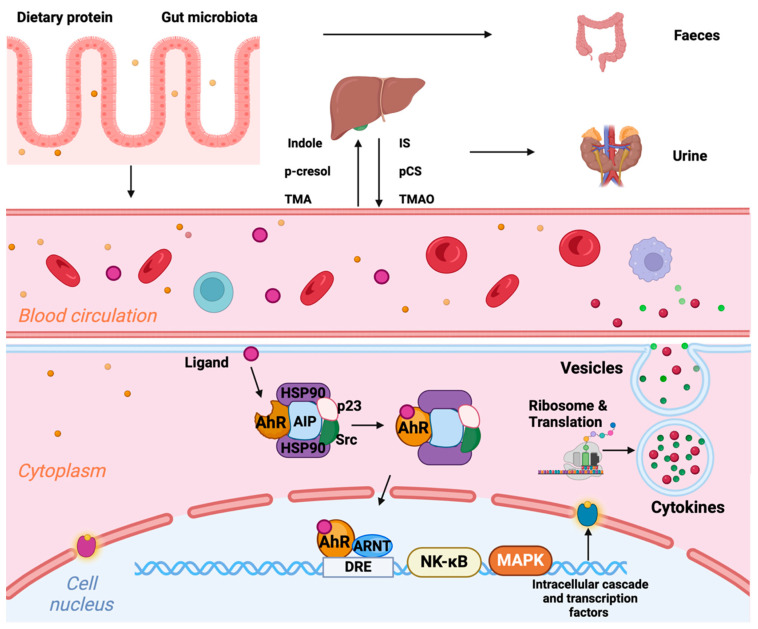
Gut-derived metabolites and the immune response. Dietary proteins, e.g., tryptophan, tyrosine, and L-carnitine (lysine), are digested by intestinal bacteria into bioactive compounds—such as indole, p-cresol, and trimethylamine (TMA)—that can either be removed via faeces or (passively) across the intestinal barrier into the circulation. The remaining precursor molecules are metabolized by hepatic enzymes into indoxyl sulphate (IS), p-cresol sulphate (pCS), and trimethylamine-N-oxide (TMAO), which are excreted by the kidneys. In CRS, the gut microbiota is altered, resulting in increased toxin production; a “leaky” cellular barrier promotes absorption; and reduced kidney clearance promotes the retention of organic compounds. These act as ligands for the aryl hydrocarbon receptor (AhR) in cells. AhR is typically inactivated in the cytoplasm as a part of a complex with heat shock protein (HSP) 90, AhR-interacting protein (AIP), p23, and pp60 Src. Ahr can be activated by the binding of endogenous or exogenous ligands, translocating to the nucleus to interact with the AhR nuclear translocator (ARNT) and binding to dioxin-response elements (DREs), leading to the NF-κB- and MAPK-dependent production of pro-inflammatory cytokines.

### 5.3. Premature Cellular Aging

CKD is associated with premature cellular senescence [128], which is characterized by the persistence of arrested cells by the induction of cyclin-dependent kinase inhibitors p16^ink4a^ and p21^cip1^ [129,130]. Senescent cells remain metabolically active and acquire a secretory phenotype (denoted SASP), characterized by the production of pro-inflammatory cytokines [131], ligands of the Wnt–β-catenin pathway [132,133], and TGF-β [134]. These mediators exert broad autocrine and paracrine effects, inducing a cascade of aging [135,136]. The accumulation of senescent cells has been documented in CKD, including IgA nephropathy, membranous nephropathy, focal segmental glomerulosclerosis, and diabetic kidney disease [128]. SASP is also a feature of cardiomyopathy where cardiomyocyte functional decline (due to DNA damage, ER stress, and mitochondrial dysfunction) contributes to the tissue remodelling characteristic of LVH. We recently demonstrated this in a pre-clinical model of CRS [127]. Cardiac fibroblast senescence has also been shown in areas of fibrotic myocardium [137,138], but a causal and pathological role remains disputed.

### 5.4. Altered Bone Mineral Metabolism

CKD is a systemic condition impacting bone metabolism, with disruptions in calcium–phosphate balance, parathyroid hormone (PTH), and vitamin D metabolism, leading to abnormalities in bone turnover, mineralization, volume, growth, and vascular calcification [139]. Higher circulating levels of inorganic phosphate, FGF23, PTH, and calciprotein particles are recognized as critical components and effectors of CKD-induced bone mineral disorders, and they directly induce inflammatory responses, leading to malnutrition, protein-energy wasting, and CVD [140,141]. As CKD advances, endocrine changes aim to increase phosphate excretion in surviving nephrons. Elevated plasma fibroblast growth factor-23 (FGF23) is activated to counteract reduced phosphate excretion, inhibiting vitamin D synthesis and causing hypocalcaemia [142]. Hypocalcaemia triggers increased PTH secretion, contributing to enhanced phosphate excretion. FGF23, acting as a myocardial growth factor, is associated with an increased LV mass in patients with CKD [143]. PTH and FGF23 levels are powerful predictors of CV death and disease risk in patients with CKD, especially those on dialysis [144]. Low vitamin D levels are similarly linked to an increased CV risk [145].

## 6. Biomarkers in CRS (Table 1)

While there is no single mechanism responsible that accounts for cardiac and renal pathobiology, endothelial dysfunction represents a final common pathway that encompasses broad molecular dysfunction. Pre-clinical CRS is characterised by endothelial dysfunction, with limited endothelium-dependent vasodilatation in response to acetylcholine [49]. Endothelial integrity and responsiveness rely on the availability of nitric oxide, the bioactive gas synthesized constitutively by endothelial (eNOS). Indeed, the total NO is decreased in patients with CKD [146,147], which likely reflects decreased production and inactivation. Bioavailable NO is required for vasodilatation, anti-thrombotic, anti-adhesive, and anti-inflammatory effects. Asymmetric dimethyl arginine (ADMA), an amino acid of intracellular origin that is produced by protein modification, is a potent inhibitor of all NOS and can uncouple the enzyme, resulting in the production of superoxide rather than NO. ADMA infusion produces vascular injury in eNOS knockout mice, suggesting a NOS-independent mechanism of injury [148]. It accumulates in CKD [149] and independently predicts vascular [150] and cardiac remodelling [151], cardiovascular outcomes, and total mortality [152]. The chronic inflammatory nature of CKD (including dialysis) and CVD—and presumably CRS—leads to upregulated inducible NOS (iNOS) and oxidative stress, which, in turn, exacerbate endothelial dysfunction through superoxide generation.

**Table 1 ijms-25-02658-t001:** Biomarkers and potential treatments for CRS.

Clinical Marker	Pathophysiological Significance	Healthy Adult Plasma Range	Relevant Pharmacological Agents (Clinical or Experimental)	References
Asymmetric dimethyl arginine (ADMA)	Reduces the following: - NO bioavailability - Oxygen delivery - Cardiac blood flow and cardiac output	0.4–1.0 µmol/L	Antioxidants	[148,153,154]
			Estrogen	
			Vitamin A	
			ACEi	
			ARBs	
			HMG-CoA reductase inhibitors (statins)	
Endothelin-1 (ET-1)	- Increases mean arterial blood pressure - Promotes vasoconstriction and myocardial/vascular fibrosis + hypertrophy - Increases cytokine + growth factor production	0.1–5 pg/mL	ET receptor antagonists (bosentan and ambrisentan)—not beneficial in HF	[155,156,157]
			SGLT2i	
Syndecan 4 (SDC4)	Part of the endothelial glycocalyx - Regulates coagulation, cell adhesion, growth, and inflammation	5.7–16.05 ng/mL	Not tested in CV literature	[158]
N-terminal prohormone of brain natriuretic peptide (NT-proBNP)	- Inhibits RAAS, ADH, and endothelin system - Promotes renal afferent arteriolar dilatation and efferent arteriolar constriction - Stimulates natriuresis	<125 pg/mL	Nesiritide (recombinant BNP)—not beneficial in HF	[159,160,161]
			Neprilysin—sacubitril (endopeptidase degrading ANP/BNP)—beneficial with ACEi or ARB	
High-sensitivity C-reactive protein (hs-CRP)	- Pro-inflammatory + prothrombotic effects—promotes vascular dysfunction - Activates RAAS - Remodels atherosclerotic plaque - Induces oxidative stress	<3 mg/L	No specific drugs target hs-CRP	
Interluekin-6 (IL-6)	Pro-inflammatory, stimulates the following: - Endothelial activation - Vascular smooth muscle cell proliferation - Leukocyte recruitment and contributes to atherosclerotic plaque growth	<5 pg/mL	Anti-IL-6 antibody (ziltivekimab)	[69,70,162,163]
			Anti-IL-6R antibody (tocilizumab)	
			Statins	
			SGLT2i	
Tumor necrosis factor alpha (TNF-α)	Pro-inflammatory - Alters vascular tone and increases microvascular permeability	<6 pg/mL	TNF-α inhibitors - Etanercept, infliximab, adalimumab, certolizumab pegol, golimumab	[110,163]
			SGLT2i	
Monocyte chemotactic protein-1 (MCP-1) (also known as CCR2)	- Promotes monocyte, macrophage, and Treg recruitment - Facilitates angiogenesis, involved in tumour/cancer development, and is associated with autoimmune, metabolic, and cardiovascular diseases	<150 pg/mL	MCP-1 antagonist (propagermanium), anti-CCR2 antibody (MLN1202)	[164,165]
			Statins PPARα activators (fenofibrate)	
			Most data are from pre-clinical studies	
CD47	Cell-surface receptor - Regulator of phagocytosis and “marker of self” - Has roles in apoptosis, proliferation, adhesion, and migration	not measured	CD47 monoclonal antibody (magrolimab)	[166,167]
Signal regulatory protein alpha (SIRP-α)	Transmembrane protein, modulates leukocyte immune responses, e.g., adhesion, migration, and phagocytosis, CD47/SIRPα axis, “don’t eat me” signal	not measured	CD47-SIRPα/Fc fusion proteins - All investigational	

Abbreviations: ADH—antidiuretic hormone; ANP—atrial natriuretic peptide; ACEi—angiotensin-converting enzyme inhibitor; ARB—angiotensin receptor blocker; CCR2—C-C chemokine receptor 2; NO—nitric oxide; PPARαperoxisome proliferator activated receptor γ; RAAS—renin–angiotensin–aldosterone system; SGLT2i—sodium-glucose cotransporter 2 inhibitor; Treg—regulatory T lymphocytes.

Endothelial damage is exacerbated by glycocalyx shedding and leads to the production of extracellular vesicles. Glycocalyx is the carbohydrate-rich protective layer lining the endothelium containing proteoglycans and glycoproteins (endothelial component) and absorbed soluble proteins, including superoxide dismutase (plasma component). It determines endothelial permeability and blood vessel–wall interactions, and it senses shear stress (the disruption of laminar blood flow), but it can be shed through matrix metalloproteinase activity in the context of vascular disease, in addition to acute insults such as hyperglycaemia and ischemia reperfusion injury. Glycocalyx damage occurs in proportion to the stage of CKD [168], with the highest levels associated with dialysis, and accumulating uremic toxins are directly implicated [169,170]. Disrupted glycocalyx is implicated following exposure to oxidised low-density lipoproteins (oxLDL) [171] and is seen in the context of atherosclerosis [172]. The relationship between shed glycocalyx and acute decompensated or chronic heart failure has not yet been robustly established [173].

Extracellular vesicles are secreted in response to endothelial injury and vary in size from exosomes (<100 nm) to microvesicles (MVs, 100–1000 nm). They carry select molecular cargo derived from the cytoplasm (nucleotides, proteins, and hormones), bear surface receptors that identify their cellular origin, and facilitate cellular cross-talk. MVs are increased in patients undergoing dialysis (and higher still in patients with diabetes), and have been shown to independently correlate with mortality [174]. In vitro, endothelial cells exposed to indoxyl sulphate release MVs that induce the expression of adhesion molecules and miRNA, increasing p53 and NF-κB expression [175]. miRNA in circulating MVs has been shown to be altered in stable coronary artery disease [176], and the levels of circulating CD144^+^ MVs were predictive of cardiovascular events [177]. miRNAs have been proposed as crucial signalling moieties in CRS, and phenotypic differences in EV that selectively package miRNA may enable the identification of the site of origin (well reviewed in [178]). However, these biomarkers are yet to be robustly studied in suitably sized clinical trials.

## 7. Therapeutic Prospects for CRS

### Recently Established Therapies

In addition to novel approaches targeting innate immunity, several established anti-inflammatory agents may hold promise in the treatment of CRS. In CKD patients, atorvastatin treatment led to reductions in hsCRP, IL-1β, and TNF levels [179]. The observed changes did *not* align with a reduction in cholesterol, supporting the previously established notion that statins exert pleiotropic anti-oxidant and/or anti-inflammatory effects [180]. Sodium-glucose cotransporter 2 (SGLT2) inhibitors, known for conferring both nephroprotective and CV benefits in patients with CKD, irrespective of diabetic status [181], also exhibit anti-inflammatory effects. Noteworthy findings include empagliflozin’s significant reduction in macrophage IL-1β release, hinting at an inhibitory effect on the NLRP3 inflammasome [182], and canagliflozin’s decrease in the plasma levels of TNF receptor 1, IL-6, and matrix metalloproteinase 7 [163]. Additionally, finerenone [183], a selective mineralocorticoid receptor antagonist proven to reduce CKD progression and CV events in the FIDELIO-DKD trial, demonstrate diverse anti-inflammatory effects [183].

Glucagon-like peptide 1 receptor (GLP1R) agonists lower blood glucose by suppressing the secretion of glucagon, stimulating the release of insulin, lowering gastric emptying, and reducing appetite [184]. A 2021 meta-analysis of eight clinical trials that included 60,080 patients with type 2 DM showed that GLP1R agonists contributed to a 14% lower risk of major CVD outcomes (myocardial infarction, stroke, or cardiovascular death) compared with the placebo [185]. Of note, the potential benefits of GLP1R agonists in kidney disease are still under investigation.

Promising effects on hs-CRP have also been documented for lipid-lowering agents, such as bempedoic acid and ezetimibe, particularly when combined with statin therapy [186]. The strategic combination of potent lipid-lowering agents with robust inflammation inhibitors represents a pivotal avenue for future drug development with additional benefit achieved through a synergistic combination of agents or the use of bi-specific monoclonal antibodies.

## 8. Novel Treatment Approaches in CRS

Various studies employing small animal models of CVD have indicated that the inhibition of the NLRP3 inflammasome, either through genetic interventions or pharmacological approaches, resulted in reduced atherosclerosis and myocardial injury. Consequently, several NLRP3 inhibitors, including MCC950, have been developed [187], albeit with variable effects on CV risk factors and disease manifestation. A recent phase 2 trial of MCC950 in patients with rheumatoid arthritis was halted due to drug-induced hepatotoxicity [188], underscoring the challenges in translating findings from mice to humans.

The non-receptor tyrosine kinase Janus Kinase 2 (Jak2) plays a crucial role in IL-6R signalling, and somatic mutations within Jak2 contribute to CHIP-driven atherosclerosis [189]. JAK inhibitors, which are currently used in the treatment of myeloproliferative diseases, are now under evaluation for other inflammatory conditions, such as rheumatoid arthritis and inflammatory bowel disease [190]. In a murine model of Jak2V617F-induced atherosclerosis, the JAK1/JAK2 inhibitor ruxolitinib demonstrated a modest reduction in atherosclerotic lesion size but induced the formation of a more unstable plaque composition by promoting lesional necrosis [191]. This suggests that the inhibition of both JAK1 and JAK2 might impact signalling pathways that are crucial for cell survival. Several other promising anti-inflammatory treatment strategies including immune checkpoint modulation, the inhibition of NET formation, the modulation of chemokine signalling to influence leukocyte–endothelial adhesion, and the stimulation or inhibition of efferocytosis pathways are putative and attractive therapeutic opportunities (well reviewed in [192]).

## 9. Emerging Perspectives in CRS

### Efferocytosis

Efferocytosis is the effective clearance of dead cells required for the resolution of tissue damage [193], and defective efferocytosis is implicated in an expanding array of chronic inflammatory diseases including atherosclerosis and recovery from myocardial infarction [194]. Macrophage polarization is crucial in cardiac wound healing, as alternatively activated macrophages enhance efferocytosis [195] and contribute to the repair of the infarcted adult murine heart [196]. The efferocytosis of apoptotic cardiomyocytes requires Mertk [197] and Legumain [198] to resolve acute inflammation and facilitate cardiac repair after permanent coronary artery occlusion and clinically relevant myocardial reperfusion. Combined MerTK and MFG-E8 deficiency in macrophages impairs efferocytosis-linked vascular endothelial growth factor (VEGF)-A secretion, facilitating angiogenesis and cardiac repair after MI [199].

In a pressure overload-induced HF model, ICAM1-deficient mice exhibited reduced monocyte recruitment, did not show signs of cardiac fibrosis, and experienced minimal ventricular dysfunction [200], mediated by an increased expression of IL-10. These data suggest that enhanced efferocytosis by cardiac resident macrophages contributes to the protective response in HF.

Defects in signals from apoptotic cells are also identified in atherosclerosis. The interaction of the universal cell-surface receptor CD47 with macrophage-bearing signal inhibitory regulatory protein (SIRP)-a results in the initiation of the “don’t-eat-me signal” that inhibits phagocytosis. Inflammatory signalling leads to inappropriate CD47 expression in apoptotic cells with atherosclerotic lesions, rendering them resistant to “being eaten”, i.e., internalization by efferocytosis [166]. Administering an anti-CD47-blocking antibody to atheroprone mice results in smaller necrotic cores and improved lesional efferocytosis. A more recent study demonstrated that the anti-CD47 antibody treatment of *Ldlr*^−/−^ mice increases resolvin D1—a specialised pro-resolving mediator—in atherosclerotic lesions [201], which then restores the full macrophage engulfment capacity of necrotic cells. In the myocardial injury-induced generation of apoptotic cells, anti-CD47 antibody treatment post-myocardial infarction improves inflammation resolution, reduces infarct size, and preserves cardiac function [202].

The “don’t eat me signal” regulated through CD47 may be manipulated by other binding partners aside from SIRP-a. The matrix protein thrombospondin 1 (TSP1) is the high-affinity soluble ligand that binds CD47, and it is significantly upregulated on apoptotic cells compared to healthy cells [203]. CD36 on macrophages also recognizes TSP1, leading to dead cell clearance. Serum TSP1 elevation is observed in patients with CKD compared to healthy controls [204].

Heterotrimeric G proteins are required for the Kidney Injury Molecule-1 (KIM-1)-mediated clearance of renal tubular epithelial cells in an acute kidney injury (AKI) model [205], as is the junctional adhesion molecule-like protein (JAML), which mediates macrophage polarisation for injury resolution [206]. The role of efferocytosis in AKI-to-CKD transition remains poorly investigated. The potential link between efferocytosis and development of CRS may stem from pathological changes within the myocardium that provide feedback to distant organs (such as the kidney) and manifest as systemic changes. For example, CRS may present as impaired renal function post-MI, characterized by increased renal macrophage infiltration and elevated levels of TGF-β and KIM-1 associated with the onset of renal fibrosis [207,208].

## 10. Sex Differences in CRS

Female-specific changes in kidney function occur throughout life stages, and the prevalence of CKD—and its attendant complications—is higher in women [209]. Sex-based differences in immune cell function affect the development and outcomes of diseases where the immune response is relevant [210]—although this remains unproven in CRS. Renal sodium, glucose, and water transporters are effective targets for treating hypertension, diabetes, and CVD. Recent comprehensive studies in rodents have detailed sexually dimorphic patterns in both the expression and abundance of electrolyte, acid–base, water, and organic solute transporters. This includes renal solute and electrolyte co-transporters, pumps, channels, claudins, and their regulators [211]. The importance of incorporating factors related to sex (and gender) in all aspects of research is increasingly recognized. CKD’s and CVD’s pathophysiology, presentation, response to therapy, and outcomes differ by sex and gender, but these factors are often not considered in basic and clinical studies.

Reporting of sex-related variables in biomedical research remains poor [212], and a chronic underrepresentation of women in CKD [213] and CVD [214] trials has been noted. A greater inclusion of female models and the incorporation of sex as a biological variable are critical to improve the rigor and reproducibility of biomedical research, including in nephrology. In the USA, eight of ten drug withdrawals from the market between 1997 and 2000 were due to greater health risks to women than to men (https://www.gao.gov/assets/gao-01-286r.pdf) (accessed on 3 January 2024), which have been attributed to a lack of inclusion of female models in the research ecosystem leading to federal drug approval.

The developments of SGLT2 inhibitors, non-steroidal mineralocorticoid agents, and glucagon-like peptide-1 agonists represent exciting advances in therapeutic options for patients with CKD and possible CRS. However, post hoc analyses of cardiovascular trials studying these agents have suggested differing effects related to sex in terms of benefit [215], as well as a greater risk of adverse events in female participants [216]. To optimize the generalizability of the results, a participation-to-prevalence ratio (PPR, defined as the proportion of participants from a particular group in a trial divided by the proportion of individuals from the same group with the disease state in the general population) of 0.8–1.2 is recommended for target study recruitment.

## 11. Concluding Remarks

The clinical concept of CRS is well established, but our molecular understanding of the disease and therapeutic options for patients have been very limited until recently. Pre-clinical studies have improved our knowledge of the range of pathophysiological mechanisms that integrate cross-talk between multiple organ systems and contribute to the disease, particularly immune activation that drives a pro-inflammatory phenotype. Recent clinical trials that included patients with CKD have revealed the specific benefit of some pharmacologic agents. However, further studies that expedite the development and testing of novel therapeutics, as well as larger-scale trials aimed at improving cardiorenal outcomes based on the severity and type of syndrome, are required.

## Figures and Tables

**Figure 1 ijms-25-02658-f001:**
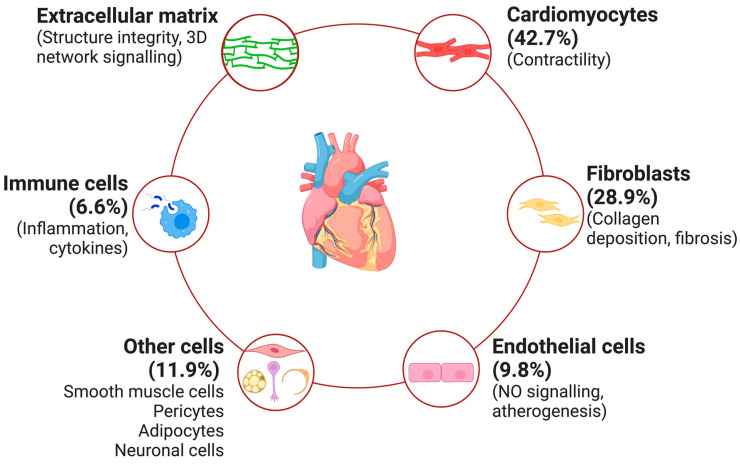
Cellular components of adult human heart. Human adult heart tissue consists of cardiomyocytes (42.7%), fibroblasts (28.9%), endothelial cells (9.8%), vascular smooth muscle cells (2.0%), pericytes (6.4%), adipocytes (3.0%), neuronal cells (0.5%), and various types of immune cells (leukocytes 6.1% and lymphocytes 0.5%). The cells are supported and segregated within a dynamic three-dimensional network—an extracellular matrix (ECM)—which is involved in regulating intercellular communication.

## Data Availability

Not applicable.

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
