# Peer review of "Novel Perspectives in Chronic Kidney Disease-Specific Cardiovascular Disease"

_ijms, 2024, doi:10.3390/ijms25052658_

Round 1

Reviewer 1 Report

Comments and Suggestions for Authors

I have a few comments, that may improve the quality of the paper:

Major comments

1. There is no clearly stated aim of the review.

2. The method of the literature search is not provided. There are 203 references and it is unclear to a reader what criteria the authors applied while selecting them. 

3. There is no conclusion. It is essential to well recapitulate such a long manuscript. This text does not give the reader a clear concluding message. The narration interrupts abruptly.

Minor comments

Page  2, line 58: the sentence" uremic toxins are not removed by dialysis" is false. They certainly are. Please clarify.

Page 7 line 253: The abbreviation TLR is introduced and it is done again on page 9, line 363.

The manuscript could be slightly better organized. Some topics are discussed repeatedly in different parts of the text. Observant reading could help.

Author Response

  1. There is no clearly stated aim of the review.

Author response: The aim of this review is clearly stated in the abstract. This review highlights the evolving paradigm and recent advances in our understanding of the molecular biology of CRS, outlining the potential for disease-specific therapies and biomarker disease detection.

  1. The method of the literature search is not provided. There are 203 references and it is unclear to a reader what criteria the authors applied while selecting them. 

Author response: As this paper is simply a review of the literature, and not a systematic review or meta-analysis per se, we have declined add further detail on the methodology used. As outlined in the Abstract and Introduction, this review is aimed at covering the most recent literature in relation to cardiorenal syndrome.

  1. There is no conclusion. It is essential to well recapitulate such a long manuscript. This text does not give the reader a clear concluding message. The narration interrupts abruptly.

Author response: We have modified the text to provide a concluding summary.

Minor comments

Page  2, line 58: the sentence" uremic toxins are not removed by dialysis" is false. They certainly are. Please clarify.

Author response: Thank you for noticing it. We have corrected the sentence adding “protein-bound”.  Protein-bound uraemic toxins (unlike water soluble toxins) are not removed by dialysis and this poses management difficulties and a substantial therapeutic challenge.  

Page 7 line 253: The abbreviation TLR is introduced and it is done again on page 9, line 363.

Author  response: We have corrected the repetition in the revised manuscript.

The manuscript could be slightly better organized. Some topics are discussed repeatedly in different parts of the text. Observant reading could help.

Author response: Thank you for this feedback. We have revised and streamlined the text.

Reviewer 2 Report

Comments and Suggestions for Authors

This fundamental review significantly underscores the importance of comprehending the molecular aspects of CRS and exploring potential therapeutic interventions, making a noteworthy contribution to the evolving paradigm in the field. I thoroughly enjoyed reading the paper and believe it is almost ready for publication. Recognizing the inherent challenge of encapsulating the entirety of such a complex topic within a single manuscript, I have only two points to highlight.

My suggestions are as follows:

In the section addressing biomarkers in CRS, I propose incorporating a table summarizing key biomarkers essential for clinical practice.

It would be beneficial to finalize the review with a dedicated concluding remarks section.  

Author Response

This fundamental review significantly underscores the importance of comprehending the molecular aspects of CRS and exploring potential therapeutic interventions, making a noteworthy contribution to the evolving paradigm in the field. I thoroughly enjoyed reading the paper and believe it is almost ready for publication. Recognizing the inherent challenge of encapsulating the entirety of such a complex topic within a single manuscript, I have only two points to highlight.

Author response: We would like to thank the reviewer for these positive comments.

My suggestions are as follows:

In the section addressing biomarkers in CRS, I propose incorporating a table summarizing key biomarkers essential for clinical practice.

It would be beneficial to finalize the review with a dedicated concluding remarks section.  

Author response: Thank you for this feedback. We have added a concluding paragraph as well as a table outlining biomarkers and relevant therapeutic options.

Reviewer 3 Report

Comments and Suggestions for Authors

Manuscript Title: Novel perspectives in CKD-specific cardiovascular disease

In the few introductory paragraphs, the authors properly contextualize the Research Topic they coordinated “Novel perspectives in CKD-specific cardiovascular disease” by recognizing CKD associated cardiovascular diseases. As correctly pointed out, authors highlight adequately recent advances in understanding molecular mechanisms behind CRS which may help in unveiling of new treatment avenues, optimize existing approaches and broaden he potential for disease-specific therapies and biomarker disease detection.

Overall, it is an interesting article for the readers. All the sections in the review are well outlined. This is a nice narrative review. The manuscript is of interest, however there are several critical omissions and areas that need improvement.

I have only the following comments for the authors which will be attractive for the reader as it will improve our general understanding on Cardio Renal Syndrome (CRS), which is a very important topic.

Abbreviations and Acronyms: Ensure that all abbreviations are defined upon initial use.

Organization and Clarity: The article is quite lengthy and dense, making it challenging for readers to follow. Consider restructuring the content and logical continuation of the text, which will improve readability.

Subsection headings should be numbered. Figure number should be removed from section/subsection heading

Figures are created from which site or software; version or licence number should be provided.

A figure should be included for better explanation of the role of gut-derived metabolites in the pathophysiology of CKD.

Reference number should be cited instead of mentioning doi or link to the article.

A table should be included for the sections “Therapeutic prospects for CRS” and “Novel Treatment Approaches”

Discussion of Limitations: When presenting potential disease specific therapeutic targets and strategies, it's essential to discuss the limitations and challenges associated with each approach.

Conclusion and Future Outlook: It should be included to offer a concise summary of key takeaways, emphasizing the importance of the discussed topics and potential clinical implications.

Comments on the Quality of English Language

Minor editing required.

Author Response

Overall, it is an interesting article for the readers. All the sections in the review are well outlined. This is a nice narrative review. The manuscript is of interest, however there are several critical omissions and areas that need improvement.

Response: We would like to thank you the reviewer for these comments.

I have only the following comments for the authors which will be attractive for the reader as it will improve our general understanding on Cardio Renal Syndrome (CRS), which is a very important topic.

Abbreviations and Acronyms: Ensure that all abbreviations are defined upon initial use.

Author response: We have corrected this at the front of the manuscript and within the text.

Organization and Clarity: The article is quite lengthy and dense, making it challenging for readers to follow. Consider restructuring the content and logical continuation of the text, which will improve readability.

Author response: We have revised and streamlined the text to improve readability.

Subsection headings should be numbered. Figure number should be removed from section/subsection heading

Figures are created from which site or software; version or licence number should be provided.

A figure should be included for better explanation of the role of gut-derived metabolites in the pathophysiology of CKD. 

Author response: We have created another figure (Figure 4) to outline the role of the gut microbiome in generating uremic toxins.

Reference number should be cited instead of mentioning doi or link to the article.

Author response: We have amended the references.

A table should be included for the sections “Therapeutic prospects for CRS” and “Novel Treatment Approaches” 

Author response: We have created Table 1 which outlines current biomarkers and relevant therapeutic options.

Discussion of Limitations: When presenting potential disease specific therapeutic targets and strategies, it's essential to discuss the limitations and challenges associated with each approach.

Author response: Thank you for this feedback. We have provided a balanced view of targets and therapeutic options as they are developed in the text, with additional information provided in Table 1.

Conclusion and Future Outlook: It should be included to offer a concise summary of key takeaways, emphasizing the importance of the discussed topics and potential clinical implications.

Author response: We have provided a concluding paragraph but have limited the text due to the already lengthy nature of the manuscript.

Round 2

Reviewer 3 Report

Comments and Suggestions for Authors

No Comments